# Characterizing the particulate content of urine in healthy humans using flow cytometry

**Sigal Hirsch[1], Ziv Porat[2], Ishai Dror[1], Yaniv Shilo[3], Brian Berkowitz[1]\***

**1** Department of Earth and Planetary Sciences, Weizmann Institute of Science, Rehovot, Israel, **2** Life Sciences Core Facilities, Weizmann Institute of Science, Rehovot, Israel, **3** Department of Urology, Kaplan Medical Center, Affiliated with the Hebrew University, Rehovot, Israel

\* brian.berkowitz@weizmann.ac.il

## Abstract

There is a notable scarcity of information concerning particulate matter in urine. This study presents an initial investigation that uses flow cytometry to determine the particulate content in the urine of healthy individuals, focusing on particles within a diameter range of 0.33–70 µm. Imaging flow cytometry was combined with fluorescent tagging and a birefringence technique to characterize particulate matter in terms of concentration, type, and size. This method enabled the identification and quantification of total particles within a sample, as well as the characterization of specific subtypes, including lipid-associated particles, protein aggregates, lipid-protein complexes, particles containing calcium (such as calcium oxalate crystals), DNA-containing particles (including cells and bacteria), and crystalline structures. Benchmark ranges for particulate matter present in urine were categorized according to subgroups that account for the influence of age, gender, and time of sampling, yielding valuable insights into the total number of particles traversing the human urinary tract daily. Significantly, the analysis here suggests that approximately $320 \times 10^6$ particles may pass through the urinary tract each day. Examination of a range of potential correlations among samples indicated that the total particle concentrations remained statistically similar. More specifically, there were no significant concentration differences in urine samples relative to sampling time, gender, or age. These findings provide valuable insights into the variability of urinary particulate matter and lay the groundwork for future, larger-scale studies. Ultimately, this research contributes to understanding urinary tract function and may potentially lead to identifying novel markers for various health conditions.

## 1. Introduction

Urine analysis for diagnostic purposes of medical conditions was first recorded by early civilizations over 6000 years ago [1,2]. However, until the 17th century, urinoscopy was based on observation of urine properties – color, character, and in some

**Data availability statement:** All relevant data are within the article and its Supporting information files.

**Funding:** This research was supported by an internal grant from the Center for Scientific Excellence, Weizmann Institute of Science.

**Competing interests:** The authors have declared that no competing interests exist.

cases taste, linked to physical symptoms such as jaundice, pain, and swelling – without chemical analysis. Urinoscopy began to evolve into urinalysis after 1637 when Thomas Brian attacked the reliance on urine "color-charts" in the "Pisse-Prophet"; this publication and many others ultimately led to the gradual introduction of chemical methods to evaluate urine. The subsequent development of increasingly sophisticated analytical methods has led to urinalysis protocols that use chemical, immunological, and microscopic techniques to examine urine and provide clinical diagnoses.

It is now recognized that human urine is composed primarily of water (95%) and is known to contain over 3000 biotic and abiotic components, in addition to urea (2%), creatinine (0.1%), and uric acid (0.03%) [2]. Urine also contains a variety of salts and ions, including microgram levels of essential and non-essential trace elements, as can be identified with techniques such as inductively coupled plasma mass spectrometry (ICP-MS) [2–4]. However, determining the precise composition of urine is complicated by fluctuations in volume and content over the course of a day, caused by a variety of parameters that differ among individuals [5,6]; even the normal range for 24-hour urine volume is between 800–2,000 mL per day, depending on a normal fluid intake of approximately 2 L per day [7]. Moreover, even under controlled conditions, urine samples can undergo compositional changes during storage, such as biogeochemical shifts and colloidal aggregation, whether stored at room temperature or under refrigeration [8,9].

It is difficult to relate urine chemistry directly – using current diagnostic tools – to all but a few specific medical conditions. Even in the context of urinary tract diseases like bladder cancer and kidney stones, several urinary markers show high sensitivity but suffer from low specificity, making them insufficient as stand-alone diagnostic tools in place of cystoscopy [10,11]. In fact, only a limited number of studies have attempted to relate urine chemistry to kidney stone formation, and significantly, (limited) comparisons of urine analyses and content between stone-forming and non-stone-forming populations have yielded no significant correlation [12–15]. Current tests for pH, uric acid, calcium, and creatinine generally provide only qualitative risk assessments rather than precise diagnostic insights. Moreover, current urinalysis offers limited information on other health conditions, such as nutritional deficiencies, autoimmune disorders, malignancies, and exposure to environmental contaminants.

In terms of particulate content, urine is known to contain various undissolved particles, including red and white blood cells, bacteria, yeast, parasites, casts, and crystals such as calcium oxalate, typically viewed under a microscope [16]. However, studies on particulate content have tended to focus on specific matters of medical (diagnostic) interest, such as markers of kidney function or malignancies; in particular, the well-established sedimentation test is an effective biomarker measure for a range of acute kidney diseases [17]. Machine learning based on image analysis of urine sediment has also been reported, although the focus was on identification and was not quantitative [18–21]. Other more general markers, such as urine microalbumin and protein content, are reported in terms of mass per volume (mg/dL), and detailed analysis of particle type, structure, and size are neglected. Consequently, the medical significance of the full particulate content in urine remains largely unknown:

its detailed analysis may lead to improved scientific and medical understanding of human body function and, potentially, to the development of diagnostic tools for various conditions.

Imaging flow cytometry is a powerful analytical technique that allows simultaneous measurement of various bright-field and fluorescence parameters on each particle, providing detailed information on particle staining intensity, distribution patterns, localization, and morphological features in liquid suspensions. This technology enables high-throughput quantification of particles on a per-milliliter basis, along with detailed characterization of particle size distributions for each particle, including its size, internal complexity, and specific markers. To date, conventional flow cytometry has been used to characterize urine particulate content with a focus on the accurate enumeration of red blood cells, leukocytes, erythrocytes, squamous epithelial cells, casts, and bacteria, as well as drug-, toxin-, and genetically-related crystalluria [22–27]; quantitative phase imaging microscopy to detect such particles has also been reported [28,29]. In particular, nephrological analyses employ urinalysis of immune and kidney parenchymal cells by flow cytometry and single-cell sequencing. For example, studies have focused on urinary T cell counts to assess kidney inflammation in rheumatic diseases and transplantation, while identification of urine tubular epithelial cells can reflect acute damage [30–32]. Recently, too, the field of urine single cell sequencing [33] is expanding with datasets on diabetes, focal segmental glomerulosclerosis, acute kidney injury and lupus nephritis [34]; imaging flow cytometry may also prove beneficial in this context.

In the present study, imaging flow cytometry was employed to more fully characterize a wide range of particulate matter in urine. By employing a combination of fluorescent dyes, the technique allowed tagging of specific components of the particles, including lipids, proteins, DNA, and calcium ions ($Ca^{2+}$), the latter of which are particularly significant as they are considered to be key contributors to the formation of calcium oxalate kidney stones. The use of these fluorescent markers enabled precise differentiation among various types of particles, enabling their classification as organic, inorganic, or mixed aggregates based on their composition. An additional capability based on birefringence was also employed to examine crystalline structures.

The primary objectives of this study were to (i) establish a rigorous methodology for the analysis of particulate matter in urine using imaging flow cytometry, in terms of concentration types and size distribution of particles and, subsequently, to (ii) characterize and quantify the particulate content (suspended matter within a diameter range of 0.33–70 µm) in the urine of apparently healthy individuals. Methodological factors influencing the results included the time of sample collection, duration time between collection and analysis, and the sample collection method. The characteristics of particulate matter across different demographic subgroups were examined, specifically comparing male and female volunteers across three age groups. These foundational steps are essential to address fundamental questions regarding the quantity and composition of particles present in urine, their size distribution, and the potential for categorizing them into groups that account for the potential influence of age and gender.

## 2. Methods and protocol development

### 2.1. Sample collection

Eighteen volunteers participated in the study, 9 females and 9 males, with three of each gender in each of three age groups ("younger"- ages 25–29 years; "middle"-ages 30–38 years; and "older"- ages 53–65 years); a nineteenth female volunteer also participated. All participants provided written informed consent, and the study received approval from the Institutional Review Board (IRB, approval #2607–1, #2567–1, #2244–2), Weizmann Institute of Science. The recruitment period for this study extended from 26 June 2023 to 3 November 2024. Urine samples were collected twice daily to account for daily fluctuations and biological variability.

Each volunteer provided sets of urine samples over three non-consecutive days. Samples were collected using standard clinical protocol, sterile urine collection cups, and sterile 10 mL vacuum tubes (MiniPlast, Israel). The samples were collected "first-in-morning" (FIM, before 9:00 AM) and "late-morning" (LM, between 10:00 AM and 12:00, the same day).

Both samples were analyzed within three hours after collection. Thus, the findings are based on 18 volunteers × 2 samples/day × 3 sampling days = 108 sample analyses.

Results of the 19th participant in the study, with unusual measurements, are discussed separately (beginning of section 3) and not included in the analysis of the other 108 samples.

## 2.2. Labeling and sample preparation

Samples were labeled with fluorescent tags to identify different potential types of particulate content. The tags are designed to label particles containing, individually or collectively, lipids (LipidTox, Thermo Fisher Scientific), protein aggregates (Proteostat, Enzo), calcium (Calcein, Sigma-Aldrich), which can be found in crystals such as calcium oxalate, and DNA (i.e., cells and bacteria; Hoechst, Thermo Fisher Scientific), as detailed in the SI, S1 Table in S1 File. Particles containing multiple components can be labeled with multiple fluorescent tags.

The samples were prepared up to three hours post-collection. Each sample was vortexed and then filtered through a 70 µm mesh to remove large particles that could clog the flow cytometer. The filtered samples were divided into smaller aliquots of 5 mL each. Subsequently, the samples were centrifuged for 5 minutes at 750 G to separate the particulate matter from the supernatant. Though biological studies typically use 300 G, a higher G force was used to maximize the particle separation.

The supernatant was discarded, and the pellet was resuspended in phosphate-buffered saline (PBS) rather than double-deionized water to prevent cell rupture or shriveling due to osmosis. The samples were then labeled with the four fluorescent tags simultaneously and incubated for the required duration (SI, S1 Table in S1 File) before undergoing second centrifugation for 5 minutes at 750 G. After discarding the supernatant, the pellet was again resuspended in 50 µL PBS (which corresponds to a 100-fold concentration from the original sample) and then measured in an imaging flow cytometer (IFC).

## 2.3. Flow cytometry method

Each sample was imaged and analyzed using an IFC (ImageStreamX Mark II, Amnis-part of Cytek Biosciences, CA, USA). Lasers were set to excite the fluorescent tags (SI, S2 Table in S1 File), with each fluorescent tag having a unique emission wavelength recorded on separate channels in the flow cytometer (SI, S3 Table in S1 File). The IFC allows for simultaneous measurements of fluorescence emissions using two detector arrays; Channels 1–6 were measured simultaneously and in parallel to channels 7–11. A minimum of 100,000 particles were collected from each sample. The instrument uses calibration beads that run along the sample. To exclude the beads, a gate was set according to the area of the bright field image and the side scatter intensity. Single stained controls were acquired to correct for inter-channel spillover, and a compensation matrix was calculated and applied to all images. The gating was done using single stained controls for each of the dyes while using the other channels as negative controls and then further verified by visual inspection. This approach was used extensively in previous unpublished work, and was adapted and validated again for this study. Once the gates were set, they were used without any change to analyze all of the samples. A schematic showing the gating strategy and definition of particle types is shown in Fig 1.

The instrument is also fitted with a custom-made setup that integrates two perpendicular polarized filters, one between the light source and the flow chamber and one before the camera [35]. This configuration restricts light transmission unless the particle alters the light polarization, as occurs with crystals, enabling the detection of crystalline materials independent of the Calcein signal, as detailed in the SI, Section S.1.1. This allowed the system to detect birefringent crystals, i.e., calcium oxalate or other mineral deposits associated with kidney stones, as well as other birefringence materials such as proteins and lipid droplets. This setup enhances the ability to visualize and quantify crystalline particles, which are often missed or underestimated in traditional urine analysis methods.

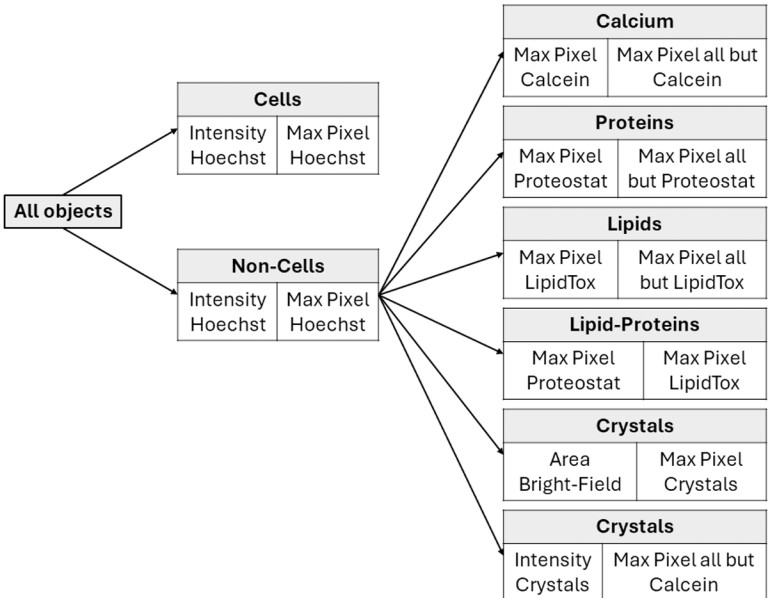

**Fig 1. Schematic illustration of the gating strategy and definition of particle types.** Each box represents the selected population and the features used to define it, as illustrated in the following plots.

## 2.4. Initial testing and particle identification

Data were analyzed using dedicated image analysis software (IDEAS 6.3; Amnis). Cells were classified first using the Hoechst signal (double-stranded DNA), while other particles were categorized as non-cells. Data were first analyzed to identify cells using a bivariate plot of intensity (total fluorescence emitted by a particle) versus max pixel (brightness of the most intense pixel) in the Hoechst channel. Particles with high intensity and max pixel values were classified as cells or cell fragments, while lower values indicated non-cells. The classification approach used to distinguish between cells and other particles is described and illustrated in the SI, Section S.1.4 and S2 Fig in S1 File.

The subsequent analysis focused on non-cell particles. Non-cell particles were then divided into four populations based on their dominant signal: Calcein-tagged ($Ca^{2+}$ ions), LipidTox-tagged (lipids), and Proteostat-tagged (protein aggregates). For each fluorescent tag, a bivariate plot of the max pixel of the specific tag was compared to the sum of the max pixel values of all other tags. This analysis allowed for the differentiation between particles tagged with one fluorescent marker, multiple markers, or no markers. The methodology is described in the SI, Section S.1.4 and S3 Fig in S1 File, using Calcein (calcium-containing particles), LipidTox (lipids), and Proteostat (protein aggregates) tags to reveal distinct populations. Analysis of Proteostat-stained particles indicated the absence of a single-stained population; instead, it indicated a mixed population with Proteostat and LipidTox tagging, indicating lipid-protein aggregates. A scatter plot of max pixel Proteostat vs. max pixel LipidTox identified three populations: those tagged predominantly with Proteostat, those with LipidTox, and those with a mixture of both (Lipid-Proteins), as shown in the SI, S4 Fig in S1 File.

Crystals were identified separately from the main populations. The identification was done by considering three complementary methods, due to the complexity of detecting particles in the polarized channel. These methods considered both birefringence signal strength and interactions with Calcein; they are described and illustrated in the SI, Section S.1.4 and S5 Fig in S1 File.

To confirm that PBS and dye aggregates do not have a significant effect on the particle measurements, the same labeling procedure was performed on samples containing PBS only (without additional filtering) instead of urine. The resulting

particle concentration was negligible, being less than 0.5% of the corresponding concentration in the urine samples in most cases, and no more than 2% in the highest case.

## 2.5. Conceptualization, initial analysis, and protocol development

In developing the protocol, several factors were considered: collection tube type, preparation time, collection time, and sample variability within a single individual and among different individuals.

The preparation time and collection tube types were initially tested on a single individual to develop a unified testing protocol. The collection tube was a 10 mL vacuum-sealed tube; this is a standardized method for handling urine samples in the medical field, and it minimizes the potential effects of air on the samples. Another collection tube was considered; for more information on the comparison, refer to the SI, Section S.1.4 and S6 Fig in S1 File.

The preparation time compared samples processed up to three hours after collection with those processed seven hours post-collection. Immediate testing was not performed for practical reasons, noting also that patient samples are not immediately analyzed in clinical settings. It was found that in a vacuum tube, the total particle concentration dropped from $250 \pm 43 \times 10^3$ particles/mL in early preparation to $101 \pm 18 \times 10^3$ particles/mL in late preparation, a 60% drop in concentration. Following that drop, a standard preparation time of three hours post-collection was implemented.

Collection time and variability were later tested among the 18 individuals. A subset of samples was selected randomly for repeated analysis to assess the natural variability of the measurement process. Of the 108 total samples, 74 samples were measured twice, irrespective of age, gender, or time of day. The results revealed an average natural variability of 30% between repeated measurements; their average was taken as the value for analysis. As detailed in Section 2.2, analyses were performed by sampling from the same 7 mL test tube and following a standardized sample preparation protocol. However, the inherently heterogeneous nature of urine introduces significant variability in key parameters, including particle morphology, size, size distribution, and composition. The sample preparation process, which involves multiple steps such as filtration, tagging, centrifugation, and resuspension, further contributes to this variability. In addition, the limitations of flow cytometry—relying on 2D imaging of large numbers (>100K) of individual particles, combined with the injection of small sample volumes (typically tens of microliters)—inherently result in some variability. Given these factors, a variability of around 30% is expected and considered acceptable, reflecting the natural inherent variation within the urine sample.

## 2.6. Statistical analysis

Statistical analyses were performed to assess differences across various groups. Two-sided t-tests were used to compare time-of-day and gender groups, while one-way analysis of variance (ANOVA) was applied to evaluate differences among age groups. The null hypothesis for each test posited that there were no significant differences between the groups being compared. A significance level of $\alpha = 0.05$ was employed, and any p-value $< 0.05$ was considered to indicate statistical significance. The results of the statistical analysis appear in the SI, Section S.8.

## 3. Results

The results shown in the following sections are based on the statistics (mean, standard deviation, or standard error) from the total of 108 samples collected from 18 volunteers. Full tables summarizing all sample analyses and statistics on natural variability appear in the SI, Sections S.2-S.9.

The analysis focused on three key factors: gender (male and female), age group (25–29 (younger), 30–38 (middle) and 53–65 (older)), and sample time (first-in-morning, FIM, and late morning, LM). These factors were examined for their impact on the composition of the particulate content. The analysis was conducted in stages: first, by grouping the results of all the volunteers to create a baseline for apparently healthy humans.

A 19th participant presented with exceptionally high particle concentrations, which resulted in a skewing of the results by approximately 25–50% (depending on the subgroup comparisons). While this could reflect natural biological variability, the

primary goal of this study was to establish a baseline for healthy individuals. To ensure that the reported results accurately reflect typical particulate content, the data from this individual were excluded from the final analyses. The data corresponding to this individual are labeled as "O1" (Outlier 1) in the SI, Sections S.3 and S.9. Given the presence of a significant birefringence signal in the samples from this participant, crystal results are addressed separately in the SI, Sections S.1.4 (especially S7 Fig in S1 File) and S.9.

### 3.1. General composition

The analysis of the samples showed that the total average concentration was $226 \pm 36 \times 10^3$ particles/mL, of which, on average, $46 \pm 5\%$ of the particles were identified. Identified particles were those tagged with at least one of LipidTox, Proteostat, and Calcein with a pixel intensity higher than 100 (arbitrary units) or Hoechst with an intensity higher than $10^5$ (arbitrary units). The remaining particles were classified as unidentified, possibly due to a lack of tagging (e.g., particles with different chemical compositions), insufficient fluorescence intensity below the cutoff threshold, clustering that obscured the fluorescence signal, and/or sample variability as discussed in Section 2.5. To increase the percentage of identified particles, future studies can utilize additional tags, as the ImageStream can currently acquire up to 10 fluorescent channels simultaneously. The mean area of total particles was $4.3 \pm 0.5$ µm$^2$; as a relative measure, this is equivalent to spherical particles (with circular areal projections) having an average diameter of 2.3 µm). The distribution of these parameters over all samples is shown in Fig 2a.

Considering that the normal range for 24-hour urine volume is 800–2,000 mL per day (an average of 1,400 mL), the estimated total particle count over 24 hours is approximately $226 \times 10^3$ particles/mL $\times$ 1,400 mL $\approx 316 \times 10^6$ particles. This calculation thus indicates that as many as 300 million suspended particles (0.33–70 µm in diameter) may pass through the urinary tract daily, with approximately 50% of these particles being identified by the current analytical protocol.

The concentrations of the individual particulate populations were as follows: lipids at $14 \pm 4 \times 10^3$ particles/mL, proteins at $30 \pm 6 \times 10^3$ particles/mL, lipid-proteins at $52 \pm 8 \times 10^3$ particles/mL, particles tagged for calcium at $4 \pm 1 \times 10^3$ particles/mL, and DNA at $1.5 \pm 1.1 \times 10^3$ particles/mL. The proportional distribution of these identified populations and the unidentified particles is depicted in Fig 2b.

Imaging flow cytometry detects particle images in terms of pixels, with the instrument used here having a minimum pixel resolution of 0.33 µm; particles are subsequently characterized in terms of area (µm$^2$) based on their imaged areal projection. The mean area of the identified populations was also measured, with lipids having a mean area of $7.0 \pm 0.4$ µm$^2$, proteins at $10.4 \pm 0.6$ µm$^2$, and lipid-proteins at $10.7 \pm 0.7$ µm$^2$, which was larger than both the proteins and lipids. Assuming circular areal projections of the particles, as a representative measure, the mean areas given here correspond to mean diameters of 3.0 µm, 3.6 µm, and 3.7 µm), respectively, for lipids, proteins, and lipid-proteins. Because the lipid-proteins were considered as aggregates of both proteins and lipids, this result was to be expected. Calcium ion particles were measured at $9.3 \pm 0.8$ µm$^2$ (diameter 3.4 µm), while DNA exhibited the largest mean area at $207 \pm 28$ µm$^2$ (diameter 16.2 µm), reflecting a mix of larger cells and smaller bacteria, as discussed in the SI, Section S.1.4. The distribution of these size measurements is shown in Fig 3. Notably, the mean areas of the tagged populations were larger than the overall average of 4.3 µm$^2$, which can be attributed to the unidentified particles, primarily consisting of much smaller particles. It is important to highlight that all mean area measurements fell within the detection limits of the instrument, with a maximum diameter of 70 µm constrained by the pipeline size and a minimum diameter of 0.33 µm determined by pixel resolution. Detailed results can be found in the SI, Section S.3. Given the limitations of the setup, providing particles sized 0.33–70 µm in diameter, the particles characterized (both the identified and the unidentified) were within the lower limit of 0.1–4900 µm$^2$.

### 3.2. Time of sampling comparison

A comparison of particle concentrations between both sample times (54 samples from each time, FIM vs. LM), without distinction based on gender or age, showed no significant variations (2-sided t-test, p = 0.95) between the different sample

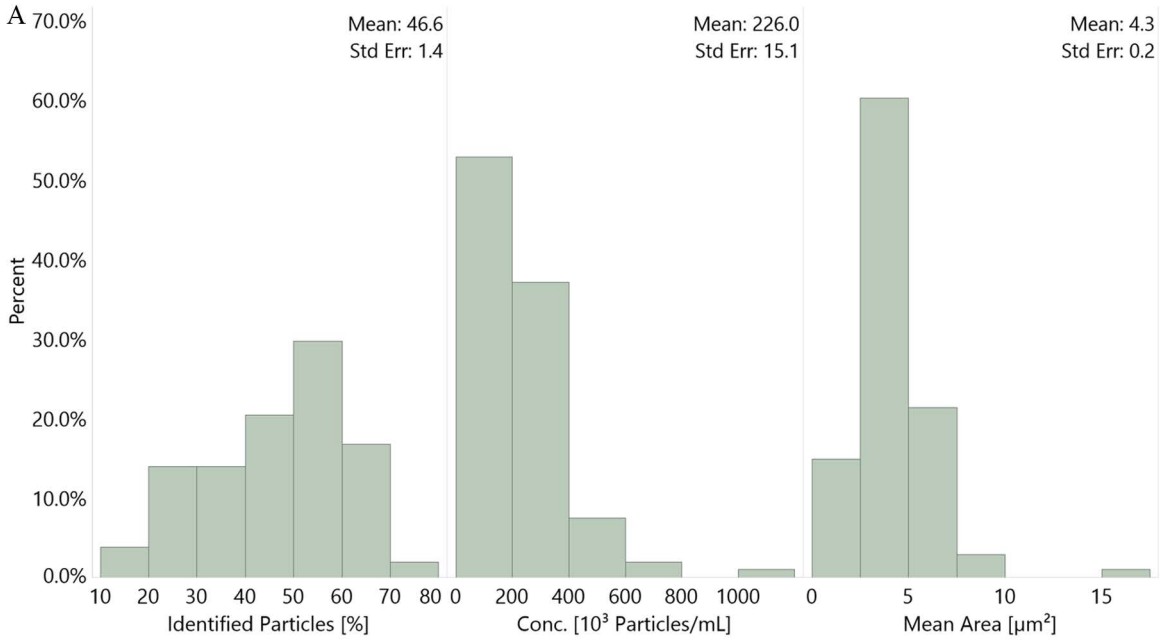

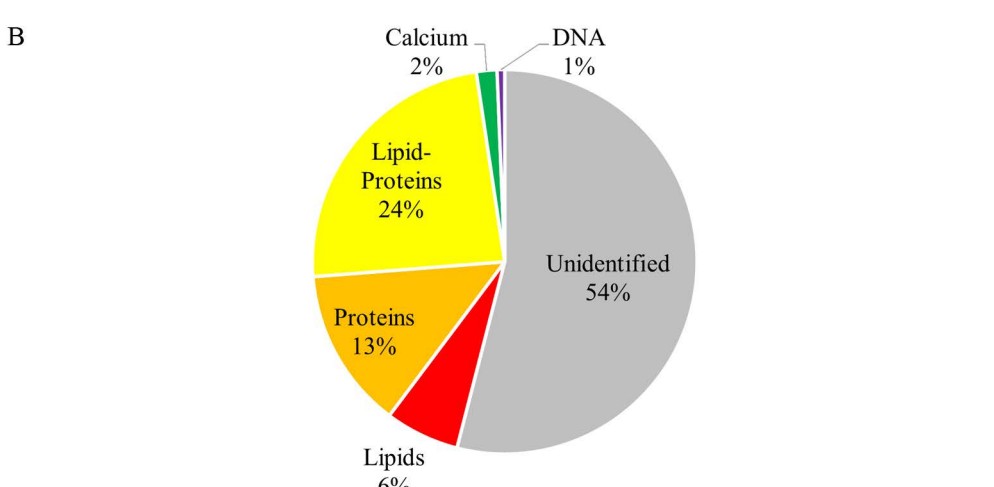

**Fig 2. Histograms and proportional distribution of particulate matter.** (a) Histogram distribution of three parameters from mean data gathered from 108 samples: frequency (percent) of each sub-range of % identified particles (left), concentration (middle), and mean area (right). The mean and standard error of each histogram appears in the top right of each block. (b) Proportional distribution of identified and unidentified particulate matter types in all the urine samples.

times. The total concentration of FIM samples was $228 \pm 39 \times 10^3$ particles/mL, while the concentration in LM samples was similar at $224 \pm 34 \times 10^3$ particles/mL. Full details are shown in the SI, Section S.4, S8 Table in S1 File.

The measurements identified $43 \pm 6\%$ of the particles in the FIM samples and $50 \pm 5\%$ of the particles in the LM samples. The difference in identification percentages was statistically significant (2-sided t-test, p = 0.01). The percentage of each subgroup among these populations is illustrated in Fig 4.

The mean area of the particles in both FIM and LM samples also showed that only the proteins had a significant difference from each other, with proteins at FIM averaging at $11.2 \pm 0.6 \; \mu m^2$ and at LM averaging at $9.6 \pm 0.6 \; \mu m^2$

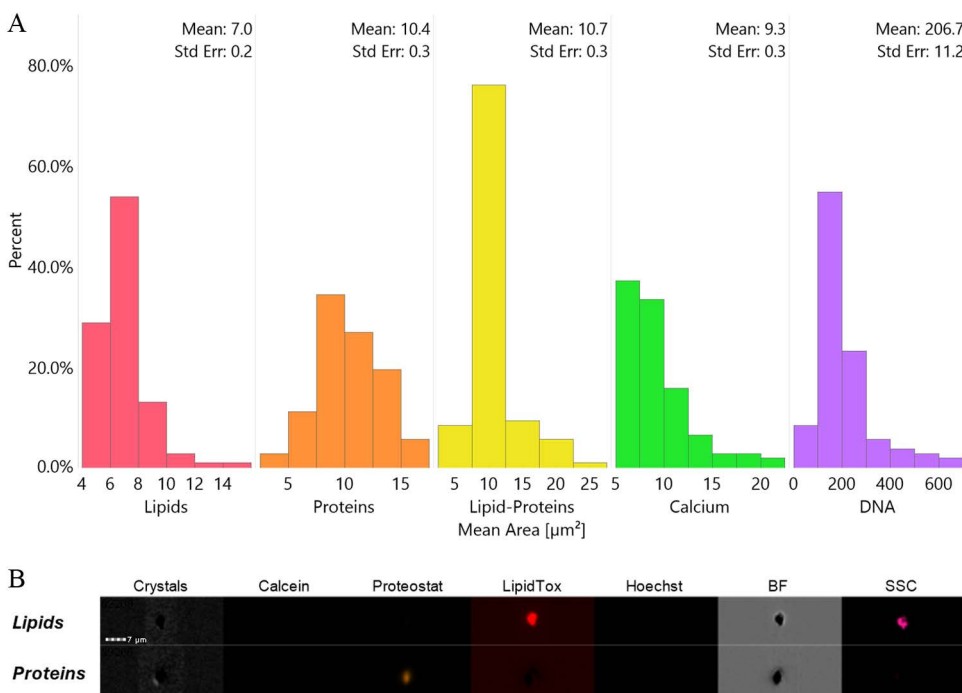

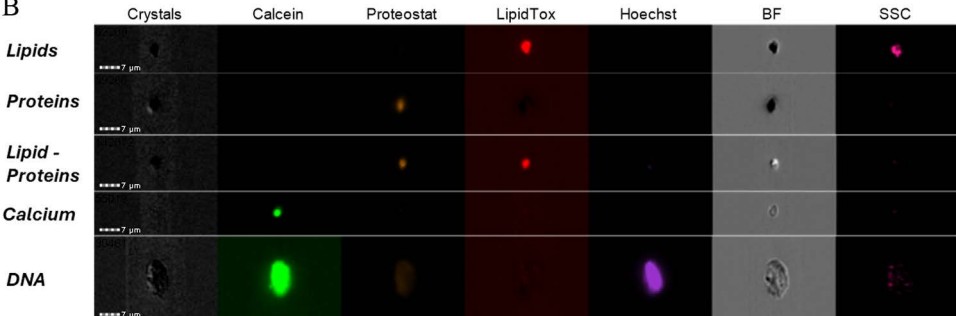

**Fig 3. (a) Histogram distribution (frequency, as percent) of mean areas of particulate content of 108 samples from the populations of lipids, proteins, lipid-proteins, particles tagged for calcium, and DNA (in order and with colors corresponding to their emission wavelength).** The mean and standard error are shown for each histogram. (b) Representative image(s) for each of the five categories of particles.

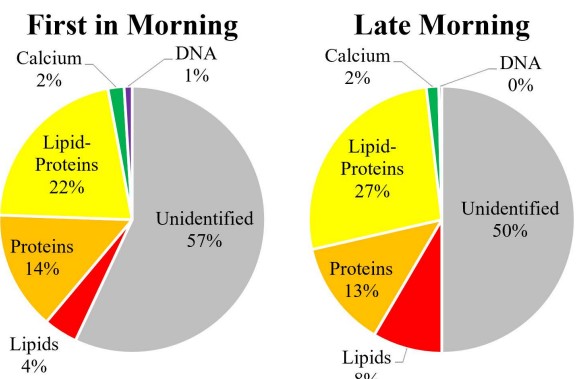

**Fig 4. Proportional distribution of identified and unidentified particulate matter types in the urine samples, 54 from each time; left: First-In-Morning (FIM) and right: Late-Morning (LM).**

(2-sided t-test, p = 0.005). All other populations had no significant difference from each other or from the combined daily average.

When examining specific populations, only lipids showed a statistically significant difference between the two sampling times (2-sided t-test, p = 0.004), with FIM samples showing a concentration of $9 \pm 1 \times 10^3$ particles/mL vs. LM samples with a concentration of $18 \pm 5 \times 10^3$ particles/mL. This translated into 4% of the particles in the FIM, increasing to 8% in the LM.

### 3.3. Gender comparison

A comparison between 9 male and 9 female volunteers (54 samples from each gender), without distinction based on age or sample time, showed no significant variations between the different genders (2-sided t-test, p = 0.695). The total particle concentration in samples from males was $222 \pm 29 \times 10^3$ particles/mL, and in samples from females, it was similar at $230 \pm 42 \times 10^3$ particles/mL. Full details may be viewed in the SI, Section S.5, S9 and S10 Tables in S1 File.

The measurements identified $40 \pm 6\%$ of the particles in samples from males, compared to $54 \pm 4\%$ in samples from females, demonstrating a statistically significant difference (2-sided t-test, p < 0.001). This significantly higher identification percentage in females is likely attributable to the presence of Hoechst-tagged (DNA) particles, which is discussed below in more detail.

The mean area of particles also differed between the genders, with only particles tagged for calcium remaining consistent between the genders. For samples from males, the general mean area was $3.2 \pm 0.3$ μm$^2$, while in samples from females, it was $5.4 \pm 0.5$ μm$^2$ (2-sided t-test, p < 0.001). With regard to specific populations, the most pronounced differences in particle size were observed in the lipid-proteins (2-sided t-test, p < 0.001) and DNA populations (2-sided t-test, p = 0.003). The protein and lipid populations were also significant (2-sided t-test, p = 0.008 and p = 0.013). The mean area of lipid-proteins particles was $9.2 \pm 0.4$ μm$^2$ in males and $12.1 \pm 0.8$ μm$^2$ in females. Similarly, DNA particles had a mean area of $174 \pm 12$ μm$^2$ in males and $239 \pm 36$ μm$^2$ in females. Full data are detailed in the SI, Section S.5.

When examining specific populations, samples from males and females showed notable differences despite the total concentration being statistically similar. Only proteins were statistically similar. The concentrations in samples from males were $19 \pm 5 \times 10^3$ particles/mL for lipids, $25 \pm 4 \times 10^3$ particles/mL for proteins, $35 \pm 5 \times 10^3$ particles/mL for lipid-proteins, and $5 \pm 1 \times 10^3$ particles/mL for particles tagged for calcium. In contrast, the concentrations in samples from females were $8 \pm 1 \times 10^3$ particles/mL for lipids, $34 \pm 7 \times 10^3$ particles/mL for proteins, $70 \pm 9 \times 10^3$ particles/mL for lipid-proteins, and $3 \pm 1 \times 10^3$ particles/mL for particles tagged for calcium. The results for the t-tests were p = 0.001 for lipids, 0.1 for proteins, < 0.001 for lipid-proteins, and 0.008 for particles tagged for calcium. The percentage of each subgroup among these populations is illustrated in Fig 5.

A significant difference was observed in the DNA particles (2-sided t-test, p = 0.005). Samples from females contained substantially more DNA particles, with a concentration of $2828 \pm 1542$ particles/mL, compared to $117 \pm 30$ particles/mL in samples from males. This difference is likely because Hoechst binds to DNA, tagging not only cells but also bacterial populations. All samples from females exhibited bacteria, which were absent in the samples from males. The bacteria are illustrated in the SI, S1 Fig in S1 File. As the volunteers were not instructed to sterilize the area before sample collection, the bacteria detected in the samples may have originated as contaminants from the skin. This difference is likely associated with the anatomical structure of the urinary tract, which varies between genders.

S2 Fig in S1 File, SI, demonstrates a clear distinction between samples from males and females, showing a noticeable bacterial population in the samples from females. Females are known to experience a higher frequency of urinary tract infections (UTIs) [36], and the presence of this bacteria—whether a contaminant or intrinsic to the sample—could provide insight into the underlying reasons for this increased susceptibility.

Interestingly, despite the presence of a measurable bacterial population in samples from females, and noting that bacteria are smaller in area than cells, the mean particle area in samples from females was still larger than in those from males. This indicates that while bacteria reduced the overall mean area, some cells in samples from females were larger than those in samples from males. The greater standard deviation in samples from females, compared to males, further

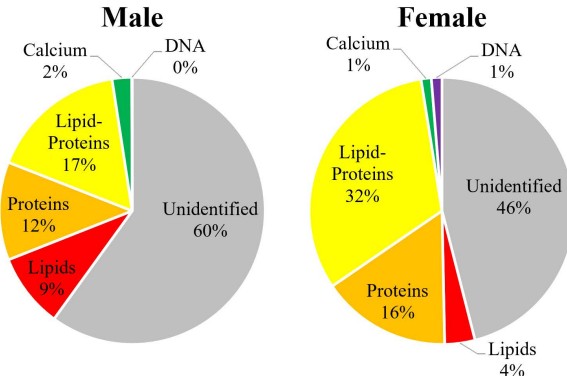

**Fig 5. Proportional distribution of identified and unidentified particulate matter types in the 54 urine samples from each of male participants (left) and female participants (right).**

suggests a wider distribution of particle sizes in females, ranging from smaller bacterial particles to larger cells. In this context, the finding that females appear to have more small DNA particles (bacteria) and more large DNA particles (cells) than males is supported by single cell studies of urine that report more pronounced shedding of transitional and squamous epithelia in female patients, which can be larger even than 70 μm [33].

### 3.4. Age group comparison

A comparison among three groups of different ages, with each group containing 6 volunteers (36 samples from each group), with no distinctions based on gender or sample collection time, indicated that there were no significant variations (one-way ANOVA, p = 0.4) among the different ages. The younger group had a concentration of $248 \pm 49 \times 10^3$ particles/mL, compared to the middle group at $226 \pm 31 \times 10^3$ particles/mL and the older group at $204 \pm 24 \times 10^3$ particles/mL. Full details appear in the SI, Section S.6, S11 and S12 Tables in S1 File.

The measurements identified $54 \pm 4\%$ in the younger group, decreased to $41 \pm 5\%$ in the middle group, and slightly increased to $44 \pm 6\%$ in the older group. The difference in identification percentages was statistically significant (one-way ANOVA, p < 0.001).

The mean particle area showed minimal variation among age groups, with the total average showing no significance between the ages. The populations that showed significant differences were lipids (one-way ANOVA, p = 0.006), lipid-proteins (one-way ANOVA, p = 0.026), and most notably particles tagged for calcium (one-way ANOVA, p < 0.001). For particles tagged for calcium, the mean area was $8.5 \pm 0.7$ μm$^2$ in participants under 50 years but significantly larger in the older group, at $10.8 \pm 0.9$ μm$^2$. This increase is likely attributable to one participant, Female 9, who had a significant concentration of crystals, which is discussed in Section 3.6.

When examining specific populations, only lipid-proteins showed a statistically significant difference between the two sampling times (one-way ANOVA, p = 0.019), with samples from the younger group showing a concentration of $68 \pm 10 \times 10^3$ particles/mL (29%), middle group with a concentration of $43 \pm 7 \times 10^3$ particles/mL (20%) and older group with a concentration of $46 \pm 6 \times 10^3$ particles/mL (23%). The proportional distribution of identified and unidentified particulate matter types across these age groups is shown in Fig 6.

### 3.5. Pair correlations

Pair correlation analysis refers to a cross-comparison between two groups, creating 4–6 subgroups for comparisons. The clear significance is highlighted in this section, and extended data for the overall and these subgroups can be found in the

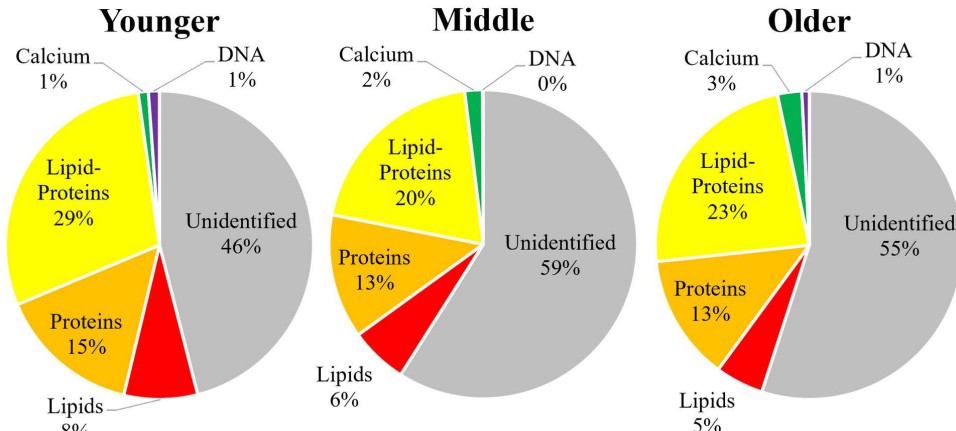

**Fig 6. Proportional distribution of identified and unidentified particulate matter types in the urine samples of different age groups (36 samples from each group): ages 25-29 years (left), ages 30-38 years (middle), and ages 53-65 years (right).**

SI, Sections S.2-S.8; a listing of the SI contents appears after the Conclusions. However, because the number of participants decreases as the number of subgroups increases, the results may be inconclusive with the smaller sample size. These subgroups could be expanded in future research.

Gender vs. Time of Day: A comparison between 9 male and 9 female volunteers, each providing samples at two time points without distinction based on age, highlighted how gender trends manifested across different sample times and, conversely, how time-of-day trends varied between genders.

Within the samples from males, there was a significant difference between the concentrations of lipids and lipid-proteins between FIM and LM samples. No significance was detected between the times of day in the samples from females. Within the FIM samples, there was a significant difference between the concentrations of lipids and DNA between the genders, with a clear significance of the concentration of lipid-proteins: $23 \pm 3 \times 10^3$ particles/mL for FIM males and $39 \pm 9 \times 10^3$ particles/mL for FIM females (2-sided t-test, $p < 0.001$). Within the LM samples, only the proteins had similar concentrations between the genders, and lipids, lipid-proteins, particles tagged for calcium, and DNA populations were significantly different.

Age Group vs. Time of Day: A comparison was made among three age groups, each consisting of 6 volunteers; each volunteer provided samples at two time points, without distinction based on gender. The analysis focused on how age trends manifested across different sample times and, conversely, how time-of-day trends varied across age groups.

No significance was detected between the times of day in the samples from the younger group. Within the middle group, there was a significant difference between the concentrations of lipids and lipid-proteins between the times of day. Within the older group, only lipids concentration varied between the times of day. Within the FIM samples, there was a significant difference between the concentrations of lipid-proteins and particles tagged for calcium between the ages. No significance was detected between the ages in the samples at LM.

Age Group vs. Gender: A comparison between three age groups with each group containing 3 male volunteers and 3 female volunteers, with no distinction based on sample collection time, highlighted how age trends manifested across different genders and, conversely, how gender trends varied across age groups.

Within the samples from males, there was a significant difference between the concentrations of lipid-proteins and particles tagged for calcium among the age groups. No significance was detected between the times of day in the samples from females. Within the younger age group, only particles tagged for calcium showed statistical similarity between the genders, while lipids, proteins, lipid-proteins, and DNA concentrations were different between the genders. This

cross-comparison is the only case in which there was any measured difference in protein concentrations, having concentrations of $21 \pm 2 \times 10^3$ particles/mL for younger males and $49 \pm 11 \times 10^3$ particles/mL for younger females (2-sided t-test, p = 0.029). Notably, only the younger group had a significant difference in DNA concentration between the genders, as the concentrations in the middle and older groups were statistically similar between the genders.

Within the middle group, there was a significant difference between the concentrations of lipids, lipid-proteins, and particles tagged for calcium between the genders. Within the older group, only the concentration of lipid-proteins showed a difference, and it was a clear significant difference: $29 \pm 3 \times 10^3$ particles/mL for older males and $63 \pm 7 \times 10^3$ particles/mL for older females (2-sided t-test, p < 0.001).

### 3.6. Crystal analysis

The analysis of crystals in urine presented challenges due to the potential variability in the form and composition of particles tagged for calcium, which do not always bind reliably to the Calcein dye. As a result, Calcein alone proved to be an unreliable marker for detecting crystals, necessitating the development of alternative methods. See SI, Sections S.1.4 and S.9 for full discussion.

The only method that achieved real crystal gating was named "Only Birefringence". The method focused on detecting particles with high intensity in the birefringence channel and minimal signal from other channels (LipidTox, Proteostat, and Hoechst), assuming that true crystals would not be tagged by non-Calcein markers.

Fig 7 shows examples of calcium oxalate dihydrate crystals that were found using the Only Birefringence method. Only one of the 18 participants showed these crystals (participant Female 19), and they appeared in two of the 6 samples. This translates to crystal abundance of 5% in humans and 2% in samples.

## 4. Discussion

Several key insights can be derived from the results presented in the previous section. First, the protocol and measurement methods delineated here demonstrate the ability to identify the presence of particulate matter in the size range of 0.33–70 μm in human urine and to estimate the concentration and many characteristics of the particles. Overall, about 50% of the particles were identified as lipid-associated particles, protein aggregates, lipid-protein complexes, particles containing calcium (such as calcium oxalate crystals), DNA-containing particles (including cells and bacteria), and crystalline structures.

Second, analysis of the total set of 108 samples indicates the presence of significant amounts of particulate matter, translating to an average of $316 \times 10^6$ particles passing through the urinary tract on a 24-hour basis. A notable finding is that the measurements showed relatively limited (statistically similar) ranges and variability in terms of types, numbers, and sizes of particles, notwithstanding the inclusion of urine samples from different genders, ages, and times of day, and without accounting for background information on lifestyle and specific medical history.

The finding of a significant amount of particulate matter is especially notable in light of the existing literature on urinalysis, which generally ignores the possible presence of particulate matter and the diagnostic information that might be

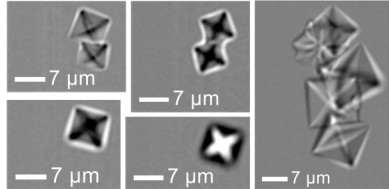

**Fig 7. Particles (calcium oxalate dihydrate crystals) found in one sample under the Only Crystal population, shown in bright field.**

gleaned from it (aside from specific analyses for detection of blood cells, malignant cells, casts, bacteria, and crystals [22–26].

Analysis of the various particle types that were identified, in terms of statistical correlations as a function of sampling time, gender, and age group, offers additional insights. Proteins were the only particle population that showed no variability between any of the correlations performed, with the exception of gender comparison of individuals of the younger group. However, because each age group had a small sample of participants per group (6 out of 18), it is possible that this result might change with larger samples. On average, the concentration of protein aggregates was $30 \times 10^3$ particles/mL, with an average size of 10 µm$^2$.

The most statistically different populations were lipids and lipid-proteins (lipids were statistically similar only in age-related comparisons, while lipid-proteins were similar only in time-related comparisons), as noted in Section 3.2. On average, the concentration of lipids was $14 \times 10^3$ particles/mL, with an average area of 7 µm$^2$, while the concentration of lipid-proteins was $52 \times 10^3$ particles/mL, with an average area of 10.7 µm$^2$. It is emphasized that lipid-proteins were identified using the lipid signal and likely contain lipids. Therefore, these results suggest that, unlike proteins, lipids vary more widely among different individuals. Moreover, lipid particle concentrations between sample times (FIM vs. LM) were statistically different, with FIM samples showing a concentration approximately half ($9 \times 10^3$ particles/mL) that of the LM samples ($18 \times 10^3$ particles/mL). This difference between time points suggests a potential connection to metabolic processes. It is plausible that the increase in lipid concentration after the first sample collection was influenced by the dietary intake of the participants. A future study could focus on monitoring changes in lipid concentrations over time, exploring whether different types of food lead to varying changes in lipid levels, or assessing the effect of a single specific food item over time to better estimate daily metabolic patterns.

DNA samples were statistically significant only in gender-related comparisons. This was attributed to bacteria, likely contamination from the skin, which is more likely to affect samples from females due to anatomical factors. Interestingly, DNA concentrations initially decreased with age but then rose again in older age groups, correlating with the prevalence of urinary tract infections [37].

The results of this study raise a key question regarding the source of these particles and, more specifically, consideration of what particles pass through the kidneys into the renal pelvis. The literature reports that particles larger than about 8 nm (not µm) are not cleared into the urine because they do not penetrate the kidney glomerular filtration barrier (GFB) [38,39]. On the other hand, there is evidence that red blood cells (~8 µm diameter; ~2 µm thickness), for example, can traverse through gaps in the glomerular capillary walls [40].

The current study identified about 50% of the particles as lipid-associated particles, protein aggregates, lipid-protein complexes, particles containing calcium (such as calcium oxalate crystals), DNA-containing particles (including cells and bacteria), and crystalline structures. The other particles were generally small (<1 µm), near the lower limit of detection of 0.33 µm. The mean size (diameter) estimated for all particles – identified and unidentified – based on an average area of $4.3 \pm 0.5$ µm$^2$ as determined by imaging flow cytometry, translates to an average particle diameter of 2.3 µm for particles with circular areal projections. However, it is important to note that the mean areas of the *identified* particle populations were significantly larger: lipids at 7.0 µm$^2$ (diameter 3.0 µm), proteins at 10.4 µm$^2$ (diameter 3.6 µm), lipid-proteins at 10.7 µm$^2$ (diameter 3.7 µm), calcium tagged particles at 9.3 µm$^2$ (diameter 3.4 µm), and DNA (larger cells and smaller bacteria) at 207 µm$^2$ (diameter 16.2 µm).

It thus remains an open question how these relatively large particles – all orders of magnitude larger than the reported 8 nm particle clearance ability of kidneys – are present in urine: do they somehow penetrate through or around the glomerular capillary walls, and/or are they formed within the urinary tract? A possible explanation is that large particles such as red blood cells and proteins are not part of kidney filtration, but are instead sloughed materials from the urothelial wall. It is well known, for example, that the occurrence of complete red blood cells in urine is indicative of post-renal sources. In addition, almost any renal cell can be found in the urine [33]. Furthermore, while it is commonly thought that normal urine

constituents usually lack lipids and proteins, it is not unusual to find small amounts of both in the urine. Indeed, millions of tubular epithelial cells may be discarded daily into the urine, and it has been suggested that the existence of lipids in urine, and more specifically among kidney stones formers, implies a relationship between lipids and stone formation [41]. Kidneys also exhibit a normal protein excretion of albumin (30%–40%) and Tamm Horsfall protein (50%) with contributions from immunoglobulins (5%–10%) and light chains (5%) [42].

While urine flow cytometry does not enable full characterization of specific urine components, exploring correlations among patients with different medical conditions may be valuable. Imaging flow cytometry excels in detecting large amounts of particles over a large size range so that, at least theoretically, it can detect prognostic significance far before it reaches clinical manifestation. For example, the detection of high levels of lipids and calcium crystals might be a prognostic tool for stone formation or recurrence. In this context, further studies with imaging flow cytometry of urine from people with known medical conditions appear justified.

Two additional aspects are noted here. First, crystals were found to be relatively rare in the urine samples, with calcium oxalate crystals detected in only 2 out of 108 samples. While there is significant literature on crystals and crystalluria (crystal presence) in urine, there appears to be little definitive quantification on the frequency of occurrence and associated concentrations. An extensive study of urine samples from 9,834 renal unit patients in a teaching hospital identified crystalluria – defined in the study as the occurrence of at least 10 crystals in 50 µL of centrifuged urine sediment, over 20 low-power microscopic fields at ×160 – in 807 (8.2%) samples [43]. Thus, the ~2% crystal occurrence identified in the current imaging flow cytometry study, from urine samples from apparently healthy volunteers, does not appear surprising. Second, the sample results from the "Outlier" (19th participant) indicate that exceptionally high particle concentrations are easily identified. Future studies based on larger sample sizes, and similar analyses of urine samples from renal colic patients, can be expected to shed light on the range of natural biological variability and the extent to which these particular results are truly an "Outlier".

## 5. Conclusions

This study represents a first quantitative and broad characterization of the suspended particle fraction of urine, recognizing the substantial data it may contain. A formal protocol that utilizes flow cytometry combined with fluorescent tagging and a birefringence technique was established to assess the particulate content (0.33 to 70 µm diameter) in the urine. This method enabled the identification and quantification of total particles within a sample, as well as the characterization of specific subtypes, including lipid-associated particles, protein aggregates, lipid-protein complexes, particles containing calcium (such as calcium oxalate crystals), DNA-containing particles (including cells and bacteria), and crystalline structures.

With this protocol, the objective was to establish benchmark ranges for the quantity and composition of particles present in urine, their size distribution, and to examine categorization according to subgroups that account for the influence of age, gender, and time of sampling.

Overall, analysis of 108 urine samples collected from 18 apparently healthy subjects, diverse in age and gender, as well as an additional 6 samples representing an "outlier" subject, offers valuable insights into the total number of particles traversing the human urinary tract daily, with additional information indicating a relatively stable composition and size over time across genders and age groups. Possibly most remarkably, the analysis here suggests that approximately $320 \times 10^6$ particles with a diameter range of 0.33–70 µm may pass through the urinary tract each day. Examination of a range of potential correlations among samples indicated that the total particle concentrations remained statistically similar. More specifically, there were no significant concentration differences in urine samples relative to sampling time, gender, or age.

This study represents an initial exploration of particulate matter in urine, and there is much to expand upon it both in terms of measurement protocols and breadth of the sample analysis. For example, other protocols, such as combining a sorter and inductively coupled plasma instrument to identify the unidentified particles recorded here, as well as particles outside the 0.33–70 µm range examined here, could also be inspected. In terms of particulate matter characterization,

it remains to investigate characteristics and correlations based on larger sample sizes and sampling that involves other external parameters (e.g., diet, exercise level, hygiene, ethnicity, family history, stress level, samples from 24-hour urine collection), as well as medical history (e.g., diabetes, bladder cancer, kidney stone, kidney function). A detailed analysis of the occurrence of crystals and structural analysis of particulate matter is another avenue for additional research.

At this stage, clearly, it is premature to attempt to link these observations to specific health conditions. However, the protocols and findings reported here show promise, especially in light of the findings from the "outlier samples" that might indicate a medical condition. Ultimately, the results reported here can be used as a baseline for future research on clinical markers and diagnosis and may ultimately contribute to the development of non-invasive diagnostic tools based on urine analysis.

## Supporting information

**S1 File.** S.1 – S.9, that include information on methods, detailed data and results, and statistical analysis. **S1 Table.** Commercial fluorescence tags that were used to label particulate matter in urine. **S2 Table.** Laser setting for the experiments in the IFC. **S3 Table.** Description of the channels and their recorded target in the IFC. **S4 Table.** Mean ± STD and range (minimum, maximum values in parentheses) particle concentration results for different collection vessels at different preparation times for 3 samples of a single donor. FIM: first-in-morning sample with preparation 3 hours post collection, LP: late-preparation 7 hours post collection, FIM + LP: daily average. **S5 Table.** Mean ± STD and range (minimum, maximum values in parentheses) particle mean area results for different collection vessels at different preparation times for single donors. FIM: first-in-morning sample with preparation 3 hours post collection, LP: late-preparation 7 hours post collection, FIM + LP: daily average. **S6 Table.** Mean ± STD and range (minimum, maximum values in parentheses) results of particle concentration for different times for 18 donors individually. FIM: first-in-morning sample, LM: late-morning samples, FIM + LM: daily average. **S7 Table.** Mean ± STD and range (minimum, maximum values in parentheses) results of particle mean area for different times for 18 donors individually. FIM: first-in-morning sample, LM: late-morning samples, FIM + LM: daily average. **S8 Table.** Mean ± SE and range (minimum, maximum values in parentheses) results for different times for 18 donors. FIM: first-in-morning sample, LM: late-morning samples, FIM + LM: daily average. **S9 Table.** Mean ± SE and range (minimum, maximum values in parentheses) results of particle concentration for different times for two gender groups, each containing 9 donors. FIM: first-in-morning sample, LM: late-morning samples, FIM + LM: daily average. **S10 Table.** Mean ± SE and range (minimum, maximum values in parentheses) results of particle mean area for different times for two gender groups, each containing 9 donors. FIM: first-in-morning sample, LM: late-morning samples, FIM + LM: daily average. **S11 Table.** Mean ± SE and range (minimum, maximum values in parentheses) results of particle concentration for different times for three age groups, each containing 6 donors. FIM: first-in-morning sample, LM: late-morning samples, FIM + LM: daily average. **S12 Table.** Mean ± SE and range (minimum, maximum values in parentheses) results of particle mean area for different times for three age groups, each containing 6 donors. FIM: first-in-morning sample, LM: late-morning samples, FIM + LM: daily average. **S13 Table.** Mean ± SE and range (minimum, maximum values in parentheses) results of particle concentration for different genders for three age groups, each containing 3 donors. FIM: first-in-morning sample, LM: late-morning samples, FIM + LM: daily average. **S14 Table.** Mean ± SE and range (minimum, maximum values in parentheses) results of particle mean area for different genders for three age groups, each containing 3 donors. FIM: first-in-morning sample, LM: late-morning samples, FIM + LM: daily average. **S15 Table.** p-values results for t-tests for time-of-day and gender comparisons, and ANOVA for age group analysis. The null hypothesis stated that there were no significant differences between the groups being compared. **S16 Table.** Mean ± STD and range (minimum, maximum values in parentheses) results of particle concentration and mean areas of 3 different crystal estimation methods for different times for 18 donors individually. FIM: first-in-morning sample, LM: late-morning samples, FIM + LM: daily average. **S2 Fig.** Population density scatter graphs of all measured objects in IFC, intensity vs. max pixel of Hoechst of (a) male, and (b) female volunteers. High values of both indicate the object measured is a

cell. The bacteria population was marked in the sample from a female. **S3 Fig.** Population density scatter graph of the measured objects in IFC of max pixel fluorescent tag of (a) LipidTox, (b) Proteostat, and (c) Calcein vs. sum of max pixel values of all other tags, taken from the "non-cells" population. Apart from Proteostat, 3 populations can be seen: only the tag, the tag, and other tags, and no tag. For Protestat, only 2 populations exist: Proteostat and LipidTox plus others. **S4 Fig.** Density population of max pixel Protestat vs. max pixel LipidTox, taken from the "non-cells" population. Three populations are shown: Proteostat tagged, LipidTox tagged, and a mix. **S5 Fig.** Three methods of finding crystals (taken from the "non-cells" population): (a) directly comparing the birefringence channel max pixel to the object size and taking the higher values of both, (b) comparing the birefringence channel max pixel to Calcein max pixel and taking the higher values of both and (c) comparing the birefringence channel max pixel to all tags but Calcein max pixel and taking the maximum of birefringence and minimum of the tags. **S6 Fig.** Histogram distribution of mean area of particulate content of 108 samples from the populations of three crystal classification. The mean and std of each histogram appear in the top right of each block. **S7 Fig.** Comparative analysis of particulate matter concentrations in urine samples of 19 participants. The three rows represent different methods of crystal estimation. Each error bar represents one standard error from the mean. F = Female volunteer, M = Male volunteer, O = Outlier, numbered by ascending age. **S8 Fig.** Population density scatter graphs of the measured objects in IFC, area vs. max pixel of particles in the birefringence gate of (a) typical sample and (b) O1 sample, taken from the "non-cells" population. Birefringence by Area population is marked on the figures. **S9 Fig.** Particles found in O1 sample under the Birefringence by Area population definition. Each row represents one object, with information about the birefringence gate signal (crystals), Calcein, Proteostat, LipidTox and Hoechst signals as well as bright field (BF) and side scatter (SSC). **S10 Fig.** Population density scatter graphs of the measured object in IFC, max pixel Calcein vs. max pixel of particles in the birefringence gate of (a) typical sample and (b) F1 sample, taken from the "non-cells" population. Birefringence by Calcein population is marked on the figures. **S11 Fig.** Particles found in O1 sample under the Birefringence by Calcein population definition. Each row represents one object, with information about the birefringence gate signal (crystals), Calcein, Proteostat, LipidTox and Hoechst signals as well as bright field (BF) and side scatter (SSC). **S12 Fig.** Population density scatter graphs of the measured object in IFC, intensity of particles in the birefringence gate vs. max pixel of LipidTox, Proteostat and Hoechst of (a) typical sample (b) F9 sample and (c) M7 sample, taken from the "non-cells" population. Only Birefringence population is marked on the figures. **S13 Fig.** Particles found in F9 sample (left) and M7 sample (right) under the Only Birefringence population, shown in bright field (BF) and birefringence gate (crystals). **S14 Fig.** Comparative analysis of particulate matter characteristics in three urine samples from the same individual based on preparation time: 3 hours (FIM, solid) vs. 7 hours (LP, textured) post-collection and sample type (vacuum, left vs. Falcon tube, right) across three replications from samples collected on different days. The three rows represent different measured parameters. The top row (Identified Particles) represents the percentage of identifiable particles. The middle row (Conc.) illustrates the total number of particles detected per milliliter in the samples. The bottom row (Mean Area) depicts the mean area of all detected particles. (DOCX)

## Acknowledgments

B.B. holds the Sam Zuckerberg Professorial Chair in Hydrology. The authors thank Dan Cojoc and two anonymous reviewers for helpful comments.

## Author contributions

**Conceptualization:** Sigal Hirsch, Ziv Porat, Ishai Dror, Yaniv Shilo, Brian Berkowitz.

**Data curation:** Sigal Hirsch, Ziv Porat, Brian Berkowitz.

**Formal analysis:** Sigal Hirsch, Ziv Porat, Ishai Dror, Yaniv Shilo, Brian Berkowitz.

**Funding acquisition:** Brian Berkowitz.

**Investigation:** Sigal Hirsch, Ziv Porat, Ishai Dror, Yaniv Shilo, Brian Berkowitz.

**Methodology:** Sigal Hirsch, Ziv Porat, Ishai Dror, Brian Berkowitz.

**Project administration:** Brian Berkowitz.

**Resources:** Brian Berkowitz.

**Software:** Sigal Hirsch, Ziv Porat.

**Supervision:** Ziv Porat, Ishai Dror, Brian Berkowitz.

**Validation:** Sigal Hirsch, Ziv Porat, Ishai Dror, Yaniv Shilo, Brian Berkowitz.

**Visualization:** Sigal Hirsch, Ziv Porat, Ishai Dror, Brian Berkowitz.

**Writing – original draft:** Sigal Hirsch, Brian Berkowitz.

**Writing – review & editing:** Sigal Hirsch, Ziv Porat, Ishai Dror, Yaniv Shilo, Brian Berkowitz.

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
