## [Decision Letter · Decision Letter 0]

27 Feb 2025

PONE-D-25-04538Characterizing the particulate content of urine in healthy humans using flow cytometryPLOS ONE

Dear Dr. Berkowitz,

Thank you for submitting your manuscript to PLOS ONE. After careful consideration, we feel that it has merit but does not fully meet PLOS ONE’s publication criteria as it currently stands. Therefore, we invite you to submit a revised version of the manuscript that addresses the points raised during the review process.

We look forward to receiving your revised manuscript.

Kind regards,

Tomasz W. Kaminski

Academic Editor

PLOS ONE

Journal Requirements:

“This research was supported by an internal grant from the Center for Scientific Excellence, Weizmann Institute of Science.”

4. Please note that funding information should not appear in the Acknowledgments section or other areas of your manuscript. We will only publish funding information present in the Funding Statement section of the online submission form. Please remove any funding-related text from the manuscript. 

Reviewers' comments:

Reviewer's Responses to Questions

**Comments to the Author**

1. Is the manuscript technically sound, and do the data support the conclusions?

Reviewer #1: Yes

Reviewer #2: Yes

Reviewer #3: Yes

2. Has the statistical analysis been performed appropriately and rigorously? 

Reviewer #1: Yes

Reviewer #2: Yes

Reviewer #3: Yes

3. Have the authors made all data underlying the findings in their manuscript fully available?

Reviewer #1: Yes

Reviewer #2: Yes

Reviewer #3: Yes

4. Is the manuscript presented in an intelligible fashion and written in standard English?

Reviewer #1: Yes

Reviewer #2: Yes

Reviewer #3: Yes

5. Review Comments to the Author

Reviewer #1: In the present manuscript Hirsch et al describe a new approach in cataloguing particulate matter in urine samples using Imaging flow cytometry. Though the donor population is admittedly small (9/9 f/m), they present a thorough analysis of multiple urine samples per donor, accounting for sampling time, age and gender. The distinction between lipids, proteins, crystals and DNA-yielding particles is very basic, but it lays the necessary groundwork for larger studies including more healthy subjects as well as patients with kidney or urinary tract diseases. It also may help in identifying confounding factors of urinalysis, thus possibly enable standardization of urine biomarkers in the future.

Overall, I think this manuscript is fit for publication with minor revisions:

- Line 92ff : In the field of nephrology, urinalysis of immune and kidney parenchymal cells by flow cytometry and single-cell sequencing has been discussed a lot more than you describe in your Introduction. There are numerous papers on urinary T cell counts reflecting kidney inflammation in rheumatic diseases and transplantation as well as some data on urine tubular epithelial cells reflecting acute damage. Also, recently the field of urine single cell sequencing has been expanding with datasets on diabetes, fsgs, acute kidney injury and lupus nephritis. Perhaps you can reflect that this is an emerging field which can benefit from your own findings.

- Line 221f : Variability of repeated measurements is 30% - that is quite a lot! Can you reflect on whether you think this is a technical or biological issue (or both?)

- Line 233ff : Should this section rather be in Methods under a statistics subheader?

- Line 270 : 50% of particles are unidentified: What do you think are these particles? How do you want to improve your measurements in the future to detect more?

- Line 370ff : the yield discrepancy of DNA particles between f and m is attributed to the occurrence of bacteriuria in females, which makes sense. However, as you state, the deviation in particle size is also much higher in females, meaning females have both more small DNA particles (bacteria) and large DNA particles (cells) – this is in line with some urine single cell papers stating a more pronounced shedding of transitional and squamous epithelia in female patients – these are large cells, sometimes even larger than your upper limit of 70µm. Maybe you can add this info as a little context.

- Line 405ff : Though I appreciate that your analysis is very thorough, from a reader’s perspective, this section drags a bit. Maybe you can shorten/ just mention significant differences / shift to supplements?

Reviewer #2: The manuscript “Characterizing the particulate content of urine in healthy humans using flow cytometry” by Hirsch et al describes in-depth analysis of human urine using imaging flow cytometry combined with multiple fluorescent probes that bind to distinct types of particulate matter. In its current state, the manuscript has a well-written introduction, providing context for the biological area. It further details methods in a thorough, direct manner. Studies are done in a rigorous manner, carefully considering variables and internal sample consistency in measurements, to provide initial benchmark values. Concerns are as follows, primarily focused on clarification of select experimental details:

1) The manuscript currently relies heavily on a large supplementary information section, with a wealth of data. The manuscript would be strengthened if some of these data were included in the main manuscript, and not supplement. For example:

i. A schematic to clarify the entire gating strategy of urine samples and how different particles were defined. This would include how beads were excluded. Currently, this information is scattered throughout supplementary data and not presented in a single, uniform figure. This is relevant, particularly as this manuscript seeks to establish methods and benchmarks for future research.

ii. Figure 2 could be further strengthened by including representative image(s) for each of the five categories (i.e. lipids, proteins, etc), to complement the quantitative data.

2) Unclear: Did the study include samples not stained with dyes, to define background fluorescence signal? If so, including data from these samples would further strengthen manuscript conclusions. If not, clarification on how populations / gates were defined should be further detailed.

3) Unclear: The parental gates for some of the plots is unclear (e.g. Fig S3, S4, S5) and should be explicitly stated in the respective figure legends.

4) In Section 3.5, manuscript would be strengthened by clarifying which specific tables contain data for each of the subheadings, given the large number of supplementary tables.

Minor details:

1) Unclear: How was the sum of the max pixels across all detectors (as in Fig S4) done?

2) Figure legends are relatively short and would be strengthened by providing further details (e.g. explicit statement of how many samples were in each comparison, even if the same between figures).

3) Fig. 1 legend would be strengthened by inclusion of a title.

4) Fig. 2 legend states mean and standard deviation are shown in Fig. 2, but plot shows standard error.

Reviewer #3: The manuscript” Characterizing the particulate content of urine in healthy humans using flow cytometry” by Sigal Hirsch et al, reports on the urinary particulate analysis by using imaging and fluorescence flow cytometry (plus birefringence for microcrystals identification).

The study is significative since it provides quantitative results of the particulates associated with the suspended particles in urine samples. The approach is original with clearly defined objective and experimental protocol, thorough analysis and presentation of the results. The paper is in general well written, with possible improvement state of the art in Introduction..

Considering the significance, novelty, methodology and quality of this study I recommend it for publication with minor revision as mentioned hereafter.

1. It is not clear why the range 0.33 – 70 microns has been chosen for particles selection. Please comment/clarify.

2. The particulate matter determined by fluorescence is associated to particle area determined by imaging, right? However, the area is determined from the image projected area for a suspended sample which, for a non-spherical, can change a lot for the same particle, because of the orientation. In principle this could influence the results. If yes, how much? Please comment/clarify.

3. Each fluorescence tag should be mentioned in methods section.

4. Flow cytometry is the main technique used to analyze urine samples and identify suspended particles. However, there are also other quantitative microscopy techniques, e.g. quantitative phase imaging, reporting detection and characterization of the particles suspended in biofluids, urine included. Some examples: Manoni, F et al, Clin. Chem. Lab. Med. 2010, 48, 1107; Ugele, M et al, Adv. Sci. 2018, 5, 1800761; Cigli, L et al, Biosensors 2023, 13, 789.

5. The distribution of the percentage (mean 46.5%) of detected particles seems is very large. Standard deviation? Could you comment on why it is so large in more in detail?

6. PLOS authors have the option to publish the peer review history of their article (what does this mean? ). If published, this will include your full peer review and any attached files.

**Do you want your identity to be public for this peer review?** For information about this choice, including consent withdrawal, please see our Privacy Policy .

Reviewer #1: No

Reviewer #2: No

Reviewer #3: **Yes: ** Dan Cojoc

---

## [Author Response · Author response to Decision Letter 0]

16 Mar 2025

Reviewers' comments:

Reviewer's Responses to Questions

Comments to the Author

1. Is the manuscript technically sound, and do the data support the conclusions?

Reviewer #1: Yes

Reviewer #2: Yes

Reviewer #3: Yes

2. Has the statistical analysis been performed appropriately and rigorously?

Reviewer #1: Yes

Reviewer #2: Yes

Reviewer #3: Yes

3. Have the authors made all data underlying the findings in their manuscript fully available?

Reviewer #1: Yes

Reviewer #2: Yes

Reviewer #3: Yes

4. Is the manuscript presented in an intelligible fashion and written in standard English?

Reviewer #1: Yes

Reviewer #2: Yes

Reviewer #3: Yes

5. Review Comments to the Author

Reviewer #1: In the present manuscript Hirsch et al describe a new approach in cataloguing particulate matter in urine samples using Imaging flow cytometry. Though the donor population is admittedly small (9/9 f/m), they present a thorough analysis of multiple urine samples per donor, accounting for sampling time, age and gender. The distinction between lipids, proteins, crystals and DNA-yielding particles is very basic, but it lays the necessary groundwork for larger studies including more healthy subjects as well as patients with kidney or urinary tract diseases. It also may help in identifying confounding factors of urinalysis, thus possibly enable standardization of urine biomarkers in the future.

Overall, I think this manuscript is fit for publication with minor revisions:

RESPONSE: We thank the referee for the positive appraisal. Below, we respond point-by-point to the suggested minor revisions, which we are pleased to incorporate.

- Line 92ff : In the field of nephrology, urinalysis of immune and kidney parenchymal cells by flow cytometry and single-cell sequencing has been discussed a lot more than you describe in your Introduction. There are numerous papers on urinary T cell counts reflecting kidney inflammation in rheumatic diseases and transplantation as well as some data on urine tubular epithelial cells reflecting acute damage. Also, recently the field of urine single cell sequencing has been expanding with datasets on diabetes, fsgs, acute kidney injury and lupus nephritis. Perhaps you can reflect that this is an emerging field which can benefit from your own findings.

RESPONSE: Thank you for providing this information. Done: As suggested, in the revised manuscript, we have expanded this paragraph to include these points and cite additional papers.

- Line 221f : Variability of repeated measurements is 30% - that is quite a lot! Can you reflect on whether you think this is a technical or biological issue (or both?)

RESPONSE: As detailed in Section 2.2 (Labeling and Sample Preparation), analyses were performed by sampling from the same 7 mL test tube and following a standardized sample preparation protocol. However, the inherently heterogeneous nature of urine introduces significant variability in key parameters, including particle morphology, size, size distribution, and composition. The sample preparation process, which involves multiple steps such as filtration, tagging, centrifugation, and resuspension, further contributes to this variability. In addition, the limitations of flow cytometry—relying on 2D imaging of large numbers (>100K) of individual particles, combined with the injection of small sample volumes (typically tens of microliters)—inherently result in some variability. Given these factors, a variability of around 30% is expected and considered acceptable, reflecting the natural inherent variation within the urine sample. Done: In the revised manuscript, we have added this additional explanation in Section 2.5, following the original Line 221 that reports the 30% value.

- Line 233ff : Should this section rather be in Methods under a statistics subheader?

RESPONSE: Agreed. Thank you for noting this. Done: In the revised manuscript, we have moved this text to a new Section 2.6.

- Line 270 : 50% of particles are unidentified: What do you think are these particles? How do you want to improve your measurements in the future to detect more?

RESPONSE: In the first paragraph of Section 3.1, where we first report that about 50% of the particles are unidentified, we noted that this is "possibly due to a lack of tagging (e.g., particles with different chemical compositions), insufficient fluorescence intensity below the cutoff threshold, and/or clustering that obscured the fluorescence signal." Done: In the revised manuscript, we have added a statement noting that "To increase the percentage of identified particles, future studies can utilize additional tags, as the ImageStream can currently acquire up to 10 fluorescent channels simultaneously."

- Line 370ff : the yield discrepancy of DNA particles between f and m is attributed to the occurrence of bacteriuria in females, which makes sense. However, as you state, the deviation in particle size is also much higher in females, meaning females have both more small DNA particles (bacteria) and large DNA particles (cells) – this is in line with some urine single cell papers stating a more pronounced shedding of transitional and squamous epithelia in female patients – these are large cells, sometimes even larger than your upper limit of 70µm. Maybe you can add this info as a little context.

RESPONSE: Agreed. Done: In the revised manuscript, we have added a statement noting that this point and adding a relevant citation.

- Line 405ff : Though I appreciate that your analysis is very thorough, from a reader’s perspective, this section drags a bit. Maybe you can shorten/ just mention significant differences / shift to supplements?

RESPONSE: We can understand the referee's comment here; we made every effort to shorten the text and mention only the key results. Indeed, we repeatedly refer the reader to the SI for detailed information, if desired. At this point, each of these subsections is essentially a concise paragraph or two, and we see no realistic way to shorten further while maintaining clarity.

Reviewer #2: The manuscript “Characterizing the particulate content of urine in healthy humans using flow cytometry” by Hirsch et al describes in-depth analysis of human urine using imaging flow cytometry combined with multiple fluorescent probes that bind to distinct types of particulate matter. In its current state, the manuscript has a well-written introduction, providing context for the biological area. It further details methods in a thorough, direct manner. Studies are done in a rigorous manner, carefully considering variables and internal sample consistency in measurements, to provide initial benchmark values. Concerns are as follows, primarily focused on clarification of select experimental details:

RESPONSE: We thank the referee for the positive appraisal. Below, we respond point by point to the suggested minor revisions, which we are pleased to incorporate.

1) The manuscript currently relies heavily on a large supplementary information section, with a wealth of data. The manuscript would be strengthened if some of these data were included in the main manuscript, and not supplement. For example:

i. A schematic to clarify the entire gating strategy of urine samples and how different particles were defined. This would include how beads were excluded. Currently, this information is scattered throughout supplementary data and not presented in a single, uniform figure. This is relevant, particularly as this manuscript seeks to establish methods and benchmarks for future research.

ii. Figure 2 could be further strengthened by including representative image(s) for each of the five categories (i.e. lipids, proteins, etc), to complement the quantitative data.

RESPONSE: Indeed, we agree that that is a huge data set that can hopefully also be used as a benchmark in future studies. We tried to balance the manuscript, providing essential information to justify the overall results and conclusion, while not wearing down the reader with the details that can be studied separately. We are pleased to accept and incorporate the reviewer's suggestions here. Done: In the revised manuscript, we have (i) added a schematic in Section 2.3 (the new Figure 1), where the gating strategy is first mentioned, to clarify the entire gating strategy of urine samples and how different particles were defined, and (ii) added to Figure 2 (now Figure 3) representative image(s) for each of the five categories of particles, to complement the quantitative data.

2) Unclear: Did the study include samples not stained with dyes, to define background fluorescence signal? If so, including data from these samples would further strengthen manuscript conclusions. If not, clarification on how populations / gates were defined should be further detailed.

RESPONSE: The gating was done using single stained controls for each of the dyes while using the other channels as negative controls, and were further verified by visual inspection. This was done extensively in previous unpublished work and was adapted and validated again for this work as well. Once the gates were set, they were used without any change for analyzing all of the samples. Done: In the revised manuscript, we have added this explanation in Section 3, first paragraph.

3) Unclear: The parental gates for some of the plots is unclear (e.g. Fig S3, S4, S5) and should be explicitly stated in the respective figure legends.

RESPONSE: The parental gate of figure S2 is all the collected particles, while all the other figures were taken from the “non-cells” gate. Done: In the revised manuscript, this is now mentioned in the text and added to the figure legends.

4) In Section 3.5, manuscript would be strengthened by clarifying which specific tables contain data for each of the subheadings, given the large number of supplementary tables.

RESPONSE: Agreed. According to the format of PLOS ONE, a listing of the SI contents is included following the Conclusions section. Done: In the revised manuscript, for clarity, we note in Section 3.5, first paragraph, that this listing appears following the Conclusions.

Minor details:

1) Unclear: How was the sum of the max pixels across all detectors (as in Fig S4) done?

RESPONSE: The Max Pixel value refers to the highest intensity pixel within the object mask, calculated for each channel. To ensure that the positive fluorescent signal emanates from the particular channel for each dye, and does not reflect mixed staining, the Max Pixel values of all the other channels were summed and plotted. For example, the “All but Calcein” parameter is calculated as: Max Pixel_M04_Proteostat + Max Pixel_M07_Hoechst + Max Pixel_M11_LipidTox. Done: In the revised SI file, we have added this explanation at the start of Section S.1. 4, where the methodology is first introduced.

2) Figure legends are relatively short and would be strengthened by providing further details (e.g. explicit statement of how many samples were in each comparison, even if the same between figures).

RESPONSE: Agreed. Done: In the revised manuscript, we have added this information in each of the figure captions.

3) Fig. 1 legend would be strengthened by inclusion of a title.

RESPONSE: Agreed. Done: In the revised manuscript, we have added a title.

4) Fig. 2 legend states mean and standard deviation are shown in Fig. 2, but plot shows standard error.

RESPONSE: Thank you for noting this typo. Done: In the revised manuscript, we have corrected the caption to read "standard error".

Reviewer #3: The manuscript” Characterizing the particulate content of urine in healthy humans using flow cytometry” by Sigal Hirsch et al, reports on the urinary particulate analysis by using imaging and fluorescence flow cytometry (plus birefringence for microcrystals identification).

The study is significative since it provides quantitative results of the particulates associated with the suspended particles in urine samples. The approach is original with clearly defined objective and experimental protocol, thorough analysis and presentation of the results. The paper is in general well written, with possible improvement state of the art in Introduction.

Considering the significance, novelty, methodology and quality of this study I recommend it for publication with minor revision as mentioned hereafter.

RESPONSE: We thank the referee for the positive appraisal. Below, we respond point by point to the suggested minor revisions, which we are pleased to incorporate.

1. It is not clear why the range 0.33 – 70 microns has been chosen for particles selection. Please comment/clarify.

RESPONSE: As noted in the original manuscript, Section 2.2 (paragraph 2), the upper limit of 70 microns is prescribed as it is necessary to remove large particles that could clog the flow cytometer. Furthermore, Section 3.1 (paragraph 4) notes that "Imaging flow cytometry detects particle images in terms of pixels, with the instrument used here having a minimum pixel resolution of 0.33 µm." This is reinforced later in the paragraph too. We believe these statements are clear and sufficient.

2. The particulate matter determined by fluorescence is associated to particle area determined by imaging, right? However, the area is determined from the image projected area for a suspended sample which, for a non-spherical, can change a lot for the same particle, because of the orientation. In principle this could influence the results. If yes, how much? Please comment/clarify.

RESPONSE: In Section 3.1 (paragraph 4) of the original manuscript, we explained that, yes, "…particles are subsequently characterized in terms of area (µm2) based on their imaged areal projection." We then stated that we assume circular areal projections of the particles. We certainly agree that for non-spherical objects, the areal projection can influence the results. However, given that the particles are imaged with random orientation, the resulting size (and thus estimated area) ranges represent an average value. Done: In the revised manuscript, we have added a caveat that we assume circular projections as a representative measure.

3. Each fluorescence tag should be mentioned in methods section.

"… tags are designed to label particles containing, individually or collectively, lipids (LipidTox, Thermo Fisher Scientific), protein aggregates (Proteostat, Enzo), calcium (Calcein, Sigma-Aldrich), which can be found in crystals such as calcium oxalate, and DNA (i.e., cells and bacteria; Hoechst, Thermo Fisher Scientific), as detailed in the SI, Table S1." We believe this text is clear and sufficient.

4. Flow cytometry is the main technique used to analyze u

---

## [Decision Letter · Decision Letter 1]

29 Mar 2025

PONE-D-25-04538R1Characterizing the particulate content of urine in healthy humans using flow cytometryPLOS ONE

Dear Dr. Berkowitz,

Thank you for submitting your manuscript to PLOS ONE. After careful consideration, we feel that it has merit but does not fully meet PLOS ONE’s publication criteria as it currently stands. Therefore, we invite you to submit a revised version of the manuscript that addresses the points raised during the review process.

We look forward to receiving your revised manuscript.

Kind regards,

Tomasz W. Kaminski

Academic Editor

PLOS ONE

Journal Requirements:

Additional Editor Comments:

Dear Authors,

Thank you for your thorough revisions. The reviewers are generally satisfied with the updates and your manuscript has been greatly improved. There is only one remaining point raised by Reviewer 2 that should still be briefly addressed. While it is a minor issue that does not significantly impact the overall paper, it would be best to discuss it and/or acknowledge it in the discussion or limitations section.

Once this final adjustment is made, the manuscript should be ready for acceptance. Please make the necessary update as soon as possible.

Best regards,

Tomasz W. Kaminski

Reviewers' comments:

Reviewer's Responses to Questions

**Comments to the Author**

1. If the authors have adequately addressed your comments raised in a previous round of review and you feel that this manuscript is now acceptable for publication, you may indicate that here to bypass the “Comments to the Author” section, enter your conflict of interest statement in the “Confidential to Editor” section, and submit your "Accept" recommendation.

Reviewer #1: All comments have been addressed

Reviewer #2: (No Response)

Reviewer #3: (No Response)

2. Is the manuscript technically sound, and do the data support the conclusions?

Reviewer #1: Yes

Reviewer #2: Yes

Reviewer #3: Yes

3. Has the statistical analysis been performed appropriately and rigorously? 

Reviewer #1: Yes

Reviewer #2: Yes

Reviewer #3: Yes

4. Have the authors made all data underlying the findings in their manuscript fully available?

Reviewer #1: Yes

Reviewer #2: Yes

Reviewer #3: Yes

5. Is the manuscript presented in an intelligible fashion and written in standard English?

Reviewer #1: (No Response)

Reviewer #2: Yes

Reviewer #3: Yes

6. Review Comments to the Author

Reviewer #1: (No Response)

Reviewer #2: In the revised manuscript “Characterizing the particulate content of urine in healthy humans using flow cytometry” by Hirsch et al, previous concerns have been appropriately addressed. On further inspection, there is one important question that remains to be addressed:

1. How did the authors address the possible contribution of particles from the diluent (i.e. PBS) or staining reagents (e.g. any of the fluorescent stains used here)? Did running PBS alone (i.e. diluent without urine) reveal any particles that may have obscured / confounded conclusions of urine particle composition? Additionally, if there were any special requirements for diluent (whether additionally filtered or obtained from a particular vendor), this information should be provided to enhance reproducibility.

Reviewer #3: (No Response)

7. PLOS authors have the option to publish the peer review history of their article (what does this mean? ). If published, this will include your full peer review and any attached files.

**Do you want your identity to be public for this peer review?** For information about this choice, including consent withdrawal, please see our Privacy Policy .

Reviewer #1: No

Reviewer #2: No

Reviewer #3: **Yes: ** Dan Cojoc

---

## [Author Response · Author response to Decision Letter 1]

18 Apr 2025

Reviewer #2: In the revised manuscript “Characterizing the particulate content of urine in healthy humans using flow cytometry” by Hirsch et al, previous concerns have been appropriately addressed. On further inspection, there is one important question that remains to be addressed:

1. How did the authors address the possible contribution of particles from the diluent (i.e. PBS) or staining reagents (e.g. any of the fluorescent stains used here)? Did running PBS alone (i.e. diluent without urine) reveal any particles that may have obscured / confounded conclusions of urine particle composition? Additionally, if there were any special requirements for diluent (whether additionally filtered or obtained from a particular vendor), this information should be provided to enhance reproducibility.

RESPONSE: A preliminary test was used to confirm that PBS and dye aggregates do not have a significant effect on the particle measurements. Potential background staining of particles originating from the PBS or dye aggregates was checked using the same labeling procedure that was performed on samples containing PBS (without additional filtering) instead of urine. The resulting particle concentration was less than 0.5% of the corresponding concentration in the urine samples in most cases, and no more than 2% in the highest case.

This information has now been included in the Revised Manuscript, at the end of Section 2.4.

As the effect of background staining was very low, we are not aware of any effect of the particular diluent on the final particle concentration or the relevance of a specific vendor.

---

## [Editor Report · Decision Letter 2]

23 Apr 2025

Characterizing the particulate content of urine in healthy humans using flow cytometry

PONE-D-25-04538R2

Dear Dr. Berkowitz,

We’re pleased to inform you that your manuscript has been judged scientifically suitable for publication and will be formally accepted for publication once it meets all outstanding technical requirements.

Kind regards,

Tomasz W. Kaminski

Academic Editor

PLOS ONE

Additional Editor Comments (optional):

All the minor issues very correctly addressed during the peer-review process. I am pleased to submit a final recommendation - Accept as it is. 
---

## [Editor Report · Acceptance letter]

PONE-D-25-04538R2

PLOS ONE

Dear Dr. Berkowitz,

I'm pleased to inform you that your manuscript has been deemed suitable for publication in PLOS ONE. Congratulations! Your manuscript is now being handed over to our production team.

Kind regards,

on behalf of

Dr. Tomasz W. Kaminski

Academic Editor

PLOS ONE